# FULL ELASTIC WEIGHT CONSOLIDATION VIA THE SURROGATE HESSIAN-VECTOR PRODUCT

## ABSTRACT

Elastic weight consolidation (EWC) is a widely accepted method for preventing catastrophic forgetting while learning a series of tasks. The key computation involved in EWC is the Fisher Information Matrix (FIM), which identifies the parameters that are crucial to previous tasks and should not be altered during new learning. However, the practical application of the FIM (a square matrix that is the same size as the number of parameters) has been limited by computational difficulties. As a result, previous uses of EWC have only employed the diagonal elements, or at most diagonal blocks, of the matrix. In this work, we introduce a method for obtaining the gradient step for EWC with the full FIM, which is both memory and computationally efficient. We evaluate the advantages of using the full FIM over just the diagonal in EWC on supervised and reinforcement learning tasks and our results demonstrate a quantitative difference between the two approaches, which are more effective when used in combination. Finally we show both empirically and theoretically that the benefits of using the full FIM are greater when the network is initialised in the lazy regime rather than the feature learning regime.

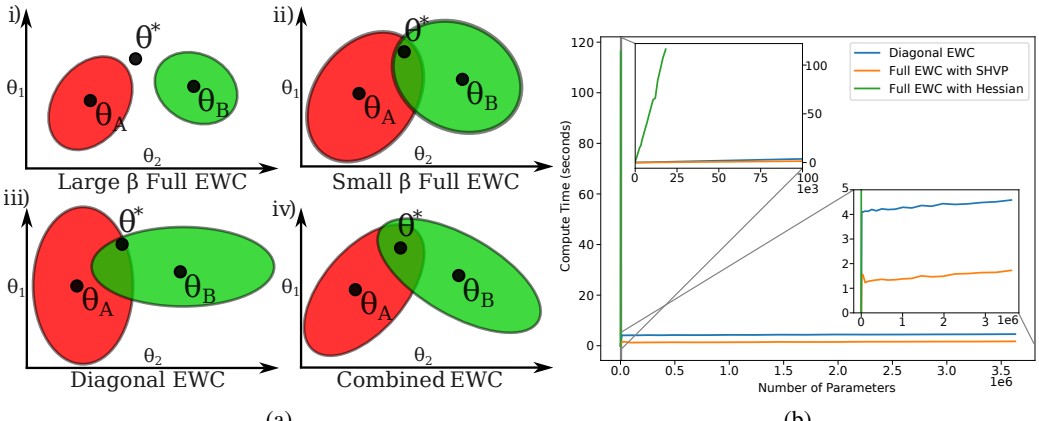

**Visual Abstract:** (a) Conceptual visualization of the kinds of Gaussian distributions fit to the neural network parameter space with the various forms of EWC and their respective benefits. $\theta_A$ is the optimal parameters for task A in isolation, $\theta_B$ is optimal for task B in isolation and $\theta^\star$ is optimal for performing both tasks concurrently. i) and ii) Full EWC has rotation but is more rounded than Diagonal EWC. Thus, there is a trade-off when working with Full EWC, either you set a large regularization rate as in (i) and risk removing the common parametrization, or set a lower regularization rate which then does not constrain the parameter space significantly as in (ii). iii) By using only the diagonal of the FIM, Diagonal EWC is limited to using axis-aligned Gaussian priors, without a rotation. This can make finding the intersection of the two task parameter distributions imprecise. iv) Finally, Combined EWC has the strengths of both diagonal and Full EWC, specifically, sharp distributions and rotation alignment. (b) Time taken to compute the various forms of EWC for increasing number of network parameters. Using the Surrogate Hessian-Vector Product (SHVP) computing the Full EWC update step is an order of magnitude ($\mathcal{O}(N)$) faster than computing the Full EWC update directly ($\mathcal{O}(N^2)$). Accounting for all computational costs we see that SHVP Full EWC is even slightly faster than Diagonal EWC.

# 1 INTRODUCTION

Elastic weight consolidation (EWC) is a continual learning method for artificial neural networks (NNs) that aims to ensure that new learning does not interfere with previously learned tasks. To achieve this, EWC uses the Fisher Information Matrix (FIM) to determine which parameters contain "information" important to previous tasks, and which can be changed freely. In this sense, parameters that are informative are merely those which have been set by prior learning. Crucially, the importance of one specific parameter is influenced by **both** the data and the other parameters of the NN. Simply put, one can only determine whether a value makes sense for one parameter when one knows the values of the other parameters in the network. Consequently, the parameters of an NN are necessarily coupled in such a manner that, when combined, they produce the correct output.

In spite of this, EWC implementations typically ignore the coupling of parameters, relying instead on only the diagonal of the FIM to limit future learning (Kirkpatrick et al., 2017). We call this approximate algorithm Diagonal EWC. Other work has used the Kronecker factored FIM which only considers the coupling of parameters in the same layer (McInerney et al., 2021; Lee et al., 2020). Fig. 1a conceptually depicts the Gaussian approximations to the loss function which correspond to each kind of EWC we cover in this work and the impact on the distribution's shape which occurs when elements of the FIM are omitted. The motivation for these approximations is predominantly computational. To consider the coupling between all parameters in an NN with $N$ parameters requires an impractical $N \times N$ FIM. In addition, to calculate the derivative of the EWC regularization term requires the multiplication of the FIM with an $N$-dimensional vector containing the deviation of the parameters from the stable point at the end of the previous tasks. The primary idea of this work is to calculate the derivative of an entirely different function than that of the EWC equation but which results in the same derivative required to perform the learning step (which we call Full EWC). Specifically, the primary contributions of this work are:

- An algorithm for computing the product of a vector and the Hessian matrix that is tractable in time and space for use in Full EWC called the *Surrogate Hessian-Vector Product*.

- A comparative study of the benefits of Full EWC compared to Diagonal EWC, which highlights complementary strengths in the two methods.

- Empirical evidence, along with theoretical justification, of a quantitative difference in the behaviour of the FIM as a function of the initialization regime (lazy or feature learning) of the network - which affects how parameters are constrained by data.

We begin by presenting the necessary background on continual learning, the FIM and EWC in Sec. 2. We then present the novel Full EWC method in Sec. 3. This section contains our primary proposition for the Surrogate Hessian-Vector Product and a comparative study with Diagonal EWC on the Permuted MNIST task. Sec. 4 then demonstrates that Full EWC provides a greater relative benefit in the lazy initialization regime and provides a theoretical explanation for this effect of initialization. Finally, in Sec. 5 we evaluate the benefit of using the full FIM on an RL domain which has been another use-case for EWC in prior work.

# 2 BACKGROUND

## 2.1 CONTINUAL LEARNING

Continual learning, sometimes called lifelong learning, is a paradigm of artificial and biological learning in which multiple tasks are learned in sequence. While definitions can vary across literature of different sub-communities, in its broadest sense it can be thought to subsume a broad class of learning problems including context detection, domain adaptation and transfer learning. Arguably the most distinct challenge associated with continual learning is so-called catastrophic forgetting (McClelland et al., 1995), in which previous information in a model is over-written by the acquisition of new information. While animals are generally very good at overcoming catastrophic forgetting, artificial neural networks trained with gradient descent methods are highly prone to interference (Goodfellow et al., 2013); this represents one of the key limitations and open problems in deep learning.

Approaches for overcoming catastrophic forgetting in neural networks broadly fall into one of three categories: dynamic architectures, replay or memory methods, and regularisation. Dynamic architecture approaches involve modifying the structure of the learning system to cope with novel tasks; for instance by increasing capacity (Rusu et al., 2016; Yoon et al., 2017), isolating and pruning/masking parameters (Mallya et al., 2018; Mallya & Lazebnik, 2018) or modularising the network (Happel & Murre, 1994; Veniat et al., 2020). Replay methods are a set of approaches inspired by the complementary learning systems (CLS) hypothesis in the cognitive science literature (McClelland et al., 1995). In CLS theory, the mammalian brain is surmised to exhibit two separate mechanisms of learning, operating on different timescales and across distinct brain regions. The first system is centred around the hippocampus and is responsible for fast learning of episodic information, while the second is distributed across cortical regions and is responsible for structured knowledge and generalisation. More generally there are two-way interactions between these systems in the form of storage, retrieval and replay; which among other proposed purposes, protects against interference. These ideas have been adopted by the machine learning community to design algorithms for combatting catastrophic forgetting by interleaving replay of previous experiences during learning of new tasks e.g. via an explicit memory buffer or by retaining a trained generative model (Shin et al., 2017). Finally, regularisation-based approaches consist of constraining learning (e.g. via a penalty term in the loss (Kirkpatrick et al., 2017; Zenke et al., 2017)) to direct learning in such a way as to avoid over-writing useful information for previous tasks. These methods have been motivated variously by information theory (Kirkpatrick et al., 2017), Bayesian theory (Farquhar & Gal, 2019) and theoretical neuroscience (Benna & Fusi, 2016). The focus of this work is elastic weight consolidation (see Sec. 2.3), which is a seminal example of this type of approach. A more comprehensive overview of the empirical literature on lifelong learning can be found in Parisi et al. (2019), Khetarpal et al. (2022), and Sodhani et al. (2022).

## 2.2 FISHER INFORMATION MATRIX

The equation for the FIM is defined as: $\quad I(\theta) = \mathbb{E}_{\mathcal{D}}[\nabla_{\theta}L(\theta)\nabla_{\theta}L(\theta)^T] \quad\quad\quad (1)$
where $\theta$ are the NN parameters, $\mathbb{E}_{\mathcal{D}}[\cdot]$ denotes the expectation operator calculated over the dataset, and $L(\theta)$ is the loss of the NN on the dataset. Naturally, this also depends on the dataset being used for the calculation of the loss: $\mathcal{D} = \{x_i, y_i\}_{i=1}^P$. As shown in Eq. 1 the FIM is the covariance of the gradients for the NN's parameters averaged over the dataset. The diagonal of the FIM contains the variance in gradients for individual parameters. Thus, even if the parameters are at a minimum in the loss landscape, with an average gradient of zero, there will still be variance in the individual parameter gradients. The values of the variance then provides a useful measure for how important a particular parameter value is. For example, if a parameter has an average gradient of $0$ but a high gradient variance then changing this parameter would result in a large increase in loss since the individual data points' losses are sensitive to this parameter value. Similarly, parameters which are not important to the learned function will not have large gradients for the individual data points and as a result lower variance. A similar point can be made for the covariance of two parameters' gradients, located on the off-diagonal of the FIM (and often omitted from the EWC regularization term). The greater the covariance of the gradients the more impact there is from changing the two parameters concurrently. Additionally, if a coordinated change in parameters results in a large increase in loss, then this is a sign that the parameters are coupled.

Secondly, it is important to note that at a minimum in the loss landscape the FIM is equal to the expected (sometimes called empirical) Hessian matrix. The equivalence of the two matrices also depends on the difference between the expected gradients and covariance of the gradients. Specifically, the empirical Hessian ($\mathbb{E}_{\mathcal{D}}[H(\theta)]$) can be factored into two additive terms: $\mathbb{E}_{\mathcal{D}}[H(\theta)] = \mathbb{E}_{\mathcal{D}}[\nabla_{\theta}L(\theta)\nabla_{\theta}L(\theta)^T] + \mathbb{E}_{\mathcal{D}}[\nabla_{\theta}L(\theta)]\mathbb{E}_{\mathcal{D}}[\nabla_{\theta}L(\theta)]^T$. However, at the minimum in the loss landscape $\mathbb{E}_{\mathcal{D}}[\nabla_{\theta}L(\theta)] = \mathbf{0}$. Consequently: $\mathbb{E}_{\mathcal{D}}[H(\theta)] = \mathbb{E}_{\mathcal{D}}[\nabla_{\theta}L(\theta)\nabla_{\theta}L(\theta)^T] = I(\theta)$. Importantly, the Hessian matrix describes the geometry of the loss landscape (Dauphin et al., 2014), justifying it and the FIM's use as metric tensors for the loss landscape (Amari, 2012). This can also be clearly shown by considering the squared norm of the loss: $||L(\theta)||^2 = L(\theta)^T L(\theta)$ (naturally the loss is a scalar value and $L(\theta)^T L(\theta) = L(\theta)^2$ however the more general case of higher-dimensional output is also valid). By replacing the loss with a first order approximation (near the minimum in the loss landscape ($\theta^*$): $L(\theta) = \nabla_{\theta}L(\theta^*)^T(\theta - \theta^*)$ and we obtain:
$||L(\theta)||^2 = (\theta - \theta^*)^T \nabla_{\theta}L(\theta^*)\nabla_{\theta}L(\theta^*)^T(\theta - \theta^*)$. Considering instead the expected loss:
$$\mathbb{E}_{\mathcal{D}}[||L(\theta)||^2] = \mathbb{E}_{\mathcal{D}}[(\theta - \theta^*)^T \nabla_{\theta}L(\theta)\nabla_{\theta}L(\theta)^T(\theta - \theta^*)]$$

$$=(\theta - \theta^*)^T \mathbb{E}_{\mathcal{D}}[\nabla_\theta L(\theta) \nabla_\theta L(\theta)^T](\theta - \theta^*)$$
$$=(\theta - \theta^*)^T I(\theta)(\theta - \theta^*)$$

If we use the minimum for a previous task $\theta^* = \theta_A$ then this is exactly the EWC regularization term as shown in Sec. 2.3. A final equivalence to note is that, in the special case of constant covariance in parameter space and independent mean parameter values, the FIM is equal to the inverse covariance matrix $I(\theta) = \Sigma^{-1}$ (Malagò & Pistone, 2015).

## 2.3 ELASTIC WEIGHT CONSOLIDATION

The supervised learning objective for training a NN $f_\theta(\cdot)$ with parameters $\theta$ on a dataset $\mathcal{D}_A = \{x_i, y_i\}_{i=1}^P$ for some task can be formulated in a Bayesian way as follows:

$$p(\theta \mid \mathcal{D}_A) = \frac{p(\mathcal{D}_A \mid \theta)p(\theta)}{p(\mathcal{D}_A)}. \tag{2}$$

Taking logs on both sides, Eq. 2 factorizes to:

$$\log(p(\theta|\mathcal{D}_A)) = \log(p(\mathcal{D}_A \mid \theta)) + \log(p(\theta)) - \log(\mathcal{D}_A) \tag{3}$$

Under the maximum a posteriori (MAP) estimation approach, we first assume a prior $p(\theta)$ on the NN parameters. If we choose a unit Gaussian prior and further assume that the likelihood $p(\mathcal{D}_A \mid \theta)$ is also Gaussian with mean $f_\theta(x)$ and identity matrix $\mathbf{I}$ as the covariance, then optimizing the RHS of Eq. 3 w.r.t. parameters $\theta$ yields the mean square error with L2-regularization. In this case, the L2 term constrains the magnitude of the parameters $\theta$. In general, this is a widely-used and successful approach for training NNs. However, this approach yields a point estimate of the optimal parameters, instead of a distribution. The point estimate $\theta_A$ that we obtain from the underlying true posterior $p(\theta|\mathcal{D}_A)$, depends on the initialization of the parameters $\theta$. Otherwise stated, there are multiple, different solutions or parameter configurations of the NN that solve task $A$ sufficiently with acceptable error. Kirkpatrick et al. (2017) refer to this phenomenon as over-parameterization and exploit it in their EWC method to achieve continual learning.

Suppose we now introduce a dataset $\mathcal{D}_B$ for a second task on which to train the same NN parameters $\theta$ that were trained on task $A$. Following the same Bayesian logic as with the objective for task $A$, we obtain:

$$\log(p(\theta|\mathcal{D}_A, \mathcal{D}_B)) = \log(p(\mathcal{D}_B \mid \mathcal{D}_A, \theta)) + \log(p(\theta \mid \mathcal{D}_A)) - \log(p(\mathcal{D}_B \mid \mathcal{D}_A)). \tag{4}$$

If we further assume that $\mathcal{D}_A, \mathcal{D}_B$ are independent, we obtain:

$$\log(p(\theta|\mathcal{D}_A, \mathcal{D}_B)) = \log(p(\mathcal{D}_B \mid \theta)) + \log(p(\theta \mid \mathcal{D}_A)) - \log(p(\mathcal{D}_B)). \tag{5}$$

Eq. 5 is almost identical to Eq. 3. The only difference between them is that we now have the posterior associated with task $A$, $p(\theta \mid \mathcal{D}_A)$ where we previously had the parameter prior distribution $p(\theta)$. In other words, the parameter posterior for task $A$ is now the prior for task $B$. The true posterior for task $A$ is intractable because the NN likelihood $p(\mathcal{D}_A \mid \theta)$ does not belong to a family of known distributions. Therefore, we need an approximation. Choosing a unit Gaussian centered at the optimized task $A$ parameters $\theta_A$ yields L2-regularization. This time, however, the L2 term penalizes large deviations from task $A$ parameters. Kirkpatrick et al. (2017) note that this approach is too conservative and fails to learn task $B$. Instead, they approximate the task $A$ posterior distribution with the Laplace approximation (MacKay, 1992), i.e., $p(\theta \mid \mathcal{D}_A) \propto \exp\left[-\frac{\lambda}{2}(\theta - \theta_A)^T I(\theta_A)(\theta - \theta_A)\right]$. This formulation is also the best-fitting Gaussian to the otherwise complicated task $A$ posterior (MacKay, 1992).

We can now approximate the RHS of Eq. 5, i.e., the loss function for fitting parameters $\theta$ on two independent tasks $A$ and $B$ jointly by:

$$\underbrace{L_{EWC}(\theta)}_{\text{joint loss}} = \underbrace{\log(p(\mathcal{D}_B \mid \theta))}_{\text{task B loss}} - \underbrace{\frac{\lambda}{2}(\theta - \theta_A)^T I(\theta_A)(\theta - \theta_A)}_{\text{regularizer}} \tag{6}$$

where we have dropped the terms which do not depend on $\theta$. Unlike L2 regularization which penalizes all weights equally, this approach only penalizes $\theta$ in regions of task $A$'s posterior that correspond to high uncertainty (Kirkpatrick et al., 2017). However, the original work only considers the diagonal FIM in favor of computational efficiency. This means that EWC can only accommodate axis-aligned approximations to the task $A$ posterior.

## 3  FULL ELASTIC WEIGHT CONSOLIDATION

The primary idea of this work is to calculate the derivative of an entirely different function than the EWC regularizer term of Eq. 6 (which we denote as $EWC(\theta)$) but which results in the same derivative. Specifically, we use Prop. 1 where $\theta_{A_{const}}$ is $\theta_A$ treated as a constant:

**Proposition 1** *Assuming that $\theta_A$ minimizes the expected loss of a prior task $\mathbb{E}_{\mathcal{D}_A}[L(\theta)]$ then:*
$$\nabla_\theta EWC(\theta) = \nabla_{\theta_A}(\theta - \theta_{A_{const}})^T \nabla_{\theta_A}\mathbb{E}_{\mathcal{D}_A}[L(\theta_A)]$$

**Proof Sketch:**  We begin by simply noting that $\nabla_\theta EWC(\theta) = (\theta - \theta_A)^T I(\theta)$. We aim to re-write this gradient step in a computationally efficient manner. To achieve this we first recognise that, at a minimum in the loss landscape, the FIM is equal to the expected Hessian matrix: $\mathbb{E}_{\mathcal{D}_A}[\nabla_{\theta_A}\nabla_{\theta_A}L(\theta_A)]$. Secondly we use the linearity of integration to change the order of expectation and differentiation: $\nabla_{\theta_A}\nabla_{\theta_A}\mathbb{E}_{\mathcal{D}_A}[L(\theta_A)]$. Finally we insert this into the EWC derivative equation and treat $(\theta - \theta_A)^T$ as constant which allows us to change the order of differentiation and multiplication from $(\theta - \theta_A)^T \nabla_{\theta_A}\nabla_{\theta_A}\mathbb{E}_{\mathcal{D}_A}[L(\theta_A)]$ to $\nabla_{\theta_A}(\theta - \theta_{A_{const}})^T \nabla_{\theta_A}\mathbb{E}_{\mathcal{D}_A}[L(\theta_A)]$.

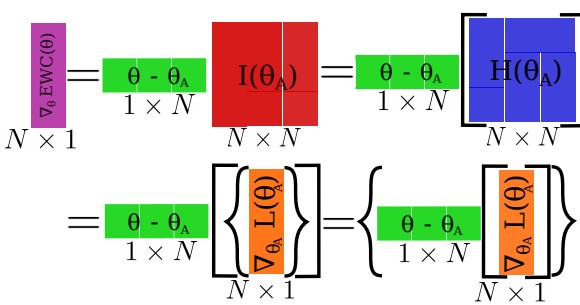

Figure 1: Visual description of the Surrogate Hessian-Vector Product. $[\cdot]$ depicts the expectation $\mathbb{E}[]$ and $\{\cdot\}$ depicts the derivative over parameters $\nabla_{\theta_A}$. The point is to take the derivative of an entirely different function to the EWC equation but results in the same derivative. Note how on the final line the product is between a $1 \times N$ and $N \times 1$ vector. Thus, we take the derivative of a scalar and never need to store an $N \times N$ matrix.

The full proof of Prop. 1 is presented in Sec. A.2, however, Figure Fig. 1 also provides a visual proof of the proposition. The main point of Prop. 1 then is that we arrive at the same derivative for EWC (which is $\mathcal{O}(N^2)$ memory and runtime complexity) by instead taking the derivative with respect to $\theta_A$ of the scalar $(\theta - \theta_{A_{const}})^T \nabla_{\theta_A}\mathbb{E}_{\mathcal{D}_A}[L(\theta_A)]$. This Surrogate Hessian-Vector Product has $\mathcal{O}(N)$ complexity which results since $(\theta - \theta_{A_{const}})$ and $\nabla_{\theta_A}\mathbb{E}_{\mathcal{D}_A}[L(\theta_A)]$ are both $N$-dimensional vectors. Thus, at no point does the full Hessian matrix need to be stored in memory.  Additionally, the Surrogate Hessian-Vector Product can be used in cases beyond just EWC, as we show in Sec. A.1, where it is applied recursively in a Neural Network Power Method to obtain the top-$k$ eigenvalues of the Hessian matrix of an NN. Other methods of computationally efficient Hessian-vector products have been proposed (Pearlmutter, 1994), however to our knowledge our exact implementation is novel and the application to EWC is new.

We also consider the combination of both the Diagonal and Full EWC terms, which we call Combined EWC. The loss function, which includes the cross entropy error term ($CE(\theta)$) and the Combined EWC regularizer, is shown in Eq. 7 with $\alpha$ and $\beta$ as regularization rates for the two EWC terms. Thus, one cost to using Combined EWC is the addition of a hyper-parameter to tune.

$$L(\theta) = CE(\theta) + \alpha \underbrace{\sum_{i=0}^{N}(\theta_i - \theta_i^*)^2 I_{ii}(\theta_A)}_{\text{Diagonal EWC}} + \beta \underbrace{\nabla_{\theta_A}(\theta - \theta_{A_{const}})^T \nabla_{\theta_A}\mathbb{E}_{\mathcal{D}_A}[L(\theta_A)]}_{\text{Full EWC using SHVP}} \qquad (7)$$

As in (Kirkpatrick et al., 2017) our first empirical results are on the permuted MNIST task. Permuted MNIST is a supervised learning task where the NN is tasked with labeling the digits presented in an input image, as in the standard MNIST task, except the pixels of the images are permuted in a consistent manner. Thus, it is possible to define multiple permuted MNIST tasks by implementing different permutations of the input pixels. In our experiments we train the NN on 5 versions of permuted MNIST in order and track the training and test accuracy of the model on all tasks after every epoch of training. In particular we are concerned with the performance of the model on the first 3 tasks as these are the most challenging to remember.

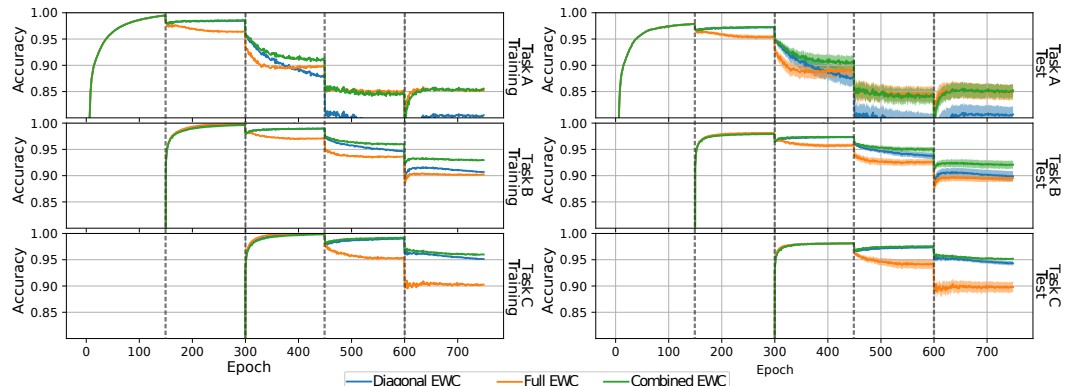

Figure 2: Training (left) and test (right) accuracies of the Diagonal, Full and Combined EWC algorithms. All algorithms are run with their optimal hyper-parameters based on validation accuracy. Error bars depict one standard deviation on either side of the mean based on 10 runs (with parameters sampled from a normal distribution with 0 mean and variance of $5 \times 10^{-3}$). Vertical lines depict task switches. Diagonal EWC has a preference towards learning new tasks and remembering recent tasks. Full EWC has the complementary strength of remembering tasks for longer but at the expense of new learning. Combined EWC outperforms both and combines their respective strengths to a degree.

We display the results of this experiment in Fig. 2. The primary observation is that at no point during the training of the five tasks does Diagonal or Full EWC out-perform Combined EWC and we see a particular improvement from Combined EWC on longer task sequences. This is clear in both figures where we see less forgetting on Task A after 4 or 5 sequences when using Combined EWC. Comparing Diagonal and Full EWC, we see that Diagonal EWC outperforms Full EWC in all cases except for Task A on the fourth and fifth tasks. This demonstrates the benefit of using the full FIM for longer task sequence lengths, as the parameter couplings of the network become more complex as more tasks are trained. Diagonal EWC, however, is better for remembering the more recent tasks. Thus Diagonal and Full EWC have complementary strengths, and when combined result in Combined EWC having the strength of both approaches.

## 4 EFFECT OF INITIALIZATION

In the permuted MNIST setting we also consider two initialization schemes, one with small initial parameter values and another with large initial values. These two initialization schemes result in the well-established regimes of training, namely the feature learning and lazy learning regimes respectively (Geiger et al., 2020). In the feature learning regime, the small initial parameters mean that the NN has to increase the magnitude of the parameter values. Identifying which parameters are necessary to increase results in the NN learning features, while unnecessary parameters remain close to 0. Alternatively, the lazy regime begins with large initial parameters which extract random features but can still be used to classify digits. Thus, in the lazy regime the parameter values tend not to change much from their initial values, giving this regime its name. The results of initializing the network in the lazy regime are shown in Fig. 3 and can be compared to the learning regime in Fig. 2 of the previous section. We see very similar results but note that the improvement afforded by Combined EWC over Diagonal EWC is more prominent in the lazy learning regime than feature learning regime. This implies that accounting for the coupling of parameters is more important in the lazy regime. In Prop. 2 we explain this phenomenon and present the full proof in Sec. A.5.

**Proposition 2** *Assuming:*

1. *$\theta_A$ minimizes the expected loss $\mathbb{E}_{D_A}[L(\theta)]$ and that $\mathbb{E}_{D_A}[L(\theta_A)] = 0$.*

2. *Constant covariance ($\Sigma(\theta) = \Sigma$) in the region of the MAP estimator ($\hat{\theta}$).*

3. *Mean parameter values which are independent of all other parameters: $\frac{\partial \mathbb{E}[\theta_i]}{\partial \theta_j} = 1$ if $i = j$ and 0 otherwise.*

*then Full EWC approximates the Bayes Optimal (Minimum Means Squared Error) estimator.*

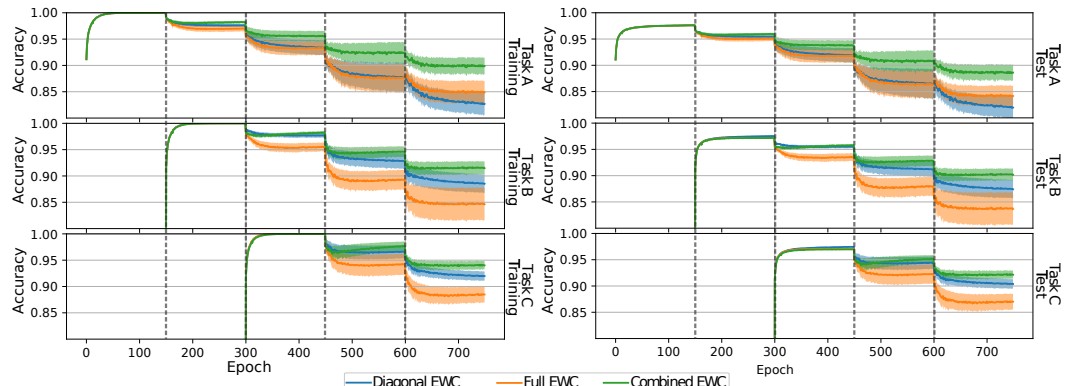

Figure 3: The effect of initialization on the relative benefit of Combined EWC: All algorithms are run with their optimal hyper-parameters based on validation accuracy. Error bars depict one standard deviation on either side of the mean based on 10 runs. We see that generalisation performance overall worsens in the large regime (with parameters sampled from a normal distribution with 0 mean and variance of 0.1). However, the relative improvement of using the full FIM in Combined EWC over the Diagonal EWC is larger.

**Proof sketch:** To prove this proposition we begin by defining a posterior distribution over the parameter space using a likelihood distribution written in exponential form and a multivariate Gaussian prior. Using this posterior distribution we obtain the expected NN function, the Bayes optimal estimator. We approximate this integral with Laplace's method around the MAP estimator. Finally, we use the special case equality of the Hessian and inverse covariance matrices to the FIM, which rely on the three assumptions, to obtain the Full EWC learning rule.

It must be noted that the three assumptions of Prop. 2 are not true in general for NNs. However, analyzing the cases where these assumptions are true and the proposition holds as a result sheds light on why Combined EWC is more beneficial in the lazy regime. Quite simply, the assumptions are true, or at least more true, in the lazy regime. This is due largely to the fact that the parameters do not move very far from there initialization in this regime and are sampled independently. It is important to note that this does not mean that the parameters are not coupled. The parameter couplings exist from the initialization of the network and learning is just able to make the most use of these couplings for the given task.

Firstly, Assumption 1 is true for both learning regimes as all tasks reach a training accuracy of 100% in the time period where that tasks is being trained (see Fig. 2 (left) and Fig. 3 (left)). Any residual loss present when the NN has reached 100% accuracy will likely be small and not result in a large difference between the FIM and Hessian. Additionally, the resulting EWC derivative will be equal. Assumption 2 effectively assumes that the loss landscape is convex around the MAP estimators. Naturally, the landscape is highly non-convex in general (Baldi & Hornik, 1989), however, in the case of large random initial parameters minor parameter changes can lead to significant changes in function space. This alone is the reason why parameters do not need to move far from their initial values to minimize the loss. Thus, in the region of the large initial parameters, and consequently the final MAP estimate, the landscape is locally convex. This is additionally the reason for the accuracy of linearized network dynamics in the Neural Tangent Kernel (Jacot et al., 2018) (the lazy regime is often also termed the NTK regime). Finally, Assumption 3 implies that, from repeated training, no parameter is consistently constrained by another parameter. In the lazy regime this is true as the initial parameters are sampled independently and the parameters do not change much. For the learning regime, however, this is not the case (especially if we account for symmetries in the network architecture). Feature learning implies that parameters of downstream layers learn to use the semantically meaningful features from the earlier layers, clearly breaking this assumption. Thus, the lazy regime appears to provide a setting where the MAP parametrization from Full EWC is Bayes optimal, explaining the relative benefits of using the full FIM in this regime over the learning regime. Taken to the infinite width limit and the NTK model, this proposition implies that only the full FIM would be required. However, in practice for NNs it appears necessary to use Combined EWC. One final point on the analysis of this section is the use of the regularization rate to constrain the prior density to a small region such that the Laplace method is a valid approximation to the

integral. As shown in Fig. 1a this runs the risk of excluding the joint optimum parametrization altogether. Thus, this analysis implies large regularization rates are used and that parameters once again do not move far from their initial values, where we can constrain the density. Next we aim to establish if a similar trend occurs in RL domains.

## 5 APPLICATION TO REINFORCEMENT LEARNING

Our final experiment aims to establish the benefits of using the full FIM is RL domains. We once again take inspiration from the original work on EWC (Kirkpatrick et al., 2017) and compare on the Atari domains (Mnih et al., 2013). In these domains the NN receives pixel inputs of the environment and must perform domain-dependent actions to achieve the highest score. For example, in Pong the agent can move north or south and is aiming to keep a ball from passing the right wall while also manoeuvring the ball past the opponent's left wall. Space Invaders however has a more complex action space with the ability to move in any cardinal direction and fire a gun. Thus, the output space of our network is set to be the size of the largest action space from any of the environments to be trained on. We train on three of the Atari domains, namely Pong, Breakout and Space Invaders, in sequence and track the performance of the model on all three domains at all timesteps. We deviate from (Kirkpatrick et al., 2017) here in that we do not show domains more than once, but train each domain for longer. Kirkpatrick et al. (2017) instead interleave shorter training sessions of each environment. Our change makes the experiment more similar to that of Sec. 3 and is less computationally intensive. Similarly, due to computational constraints, we train on less domains than Kirkpatrick et al. (2017) however our current experimental approach appears to be sufficient for determining the trade-off presented by Diagonal versus Full EWC. Finally, we us the recommended hyper-parameters for a Double-DQN (Van Hasselt et al., 2016) which includes maintaining a replay buffer. This replay buffer persists between tasks and, as mentioned in Sec. 2.1, this alone does help continual learning. This buffer is consistent between all models and so does not affect our ability to draw conclusions on their relative performance. All regularization rates were tuned for each model to maximize the standardized sum of rewards across the three environments (a measure of the model's net performance).

The results of the RL experiment are shown in Fig. 4 (right) which displays the average sum of rewards across the three tasks as learning occurs (all tasks rewards are standardized to lie between 0.0 and 1.0 based on the minimum and maximum rewards across all models). The average is taken over 500 timesteps. Thus, for an agent to increase it's score on this figure, it is not enough to avoid forgetting prior tasks, but must still allow for new learning. We once again see that the Combined EWC performs best overall, allowing for new learning and avoids forgetting prior learning. This can also be seen in Fig. 4 (left) which demonstrates the performance of each model on each environment over time. We see that Combined EWC does not completely learn Breakout after Pong but then is able to learn Space Invaders (relative to a vanilla Double DQN agent in the given time-frame). Looking at Full EWC we see that it is relatively incapable of learning new information once it has trained on Pong. Conversely Diagonal EWC forgets Pong almost immediately and prioritizes learning some new information for Breakout and Space Invaders, although with a higher degree of variance in general. Thus, we once again see the trade-off identified in Sec. 3. Diagonal EWC prioritizes recent tasks while Full EWC prioritizes the first tasks which it is trained on. Consequently, they can be combined effectively to obtain a model which remembers tasks over a long time-frame but is also capable of some new learning.

We do note that there are limitations to our current experimental setup. As mentioned in Sec. 2.3, an important assumption of EWC is that the network used is over-parametrized for performing all of the required tasks. If this is not the case then a relatively higher proportion of parameters will be useful for the first few tasks and the FIM will just identify that all parameters should be regularized to avoid forgetting. For our RL results on the Atari domains it is unlikely that our model is sufficiently over-parametrized for all three tasks. To determine if size is proving to be a limiting factor we rerun the same Atari experiment with a network with half the number of parameters in Sec. A.7.1 and indeed see a drop in performance. Thus, it is unlikely that the architecture in Sec. 5 is sufficiently over-parameterized. However, determining which version of EWC provides better performance in a parameter constrained setting is still an important experiment. The experiment does clearly demonstrates the trade-off between Diagonal versus Full EWC and the superior performance from combining the two. This could likely be due to Combined EWC being more precise in its regularization and as a result less wasteful with its parameter space and less affected by not being

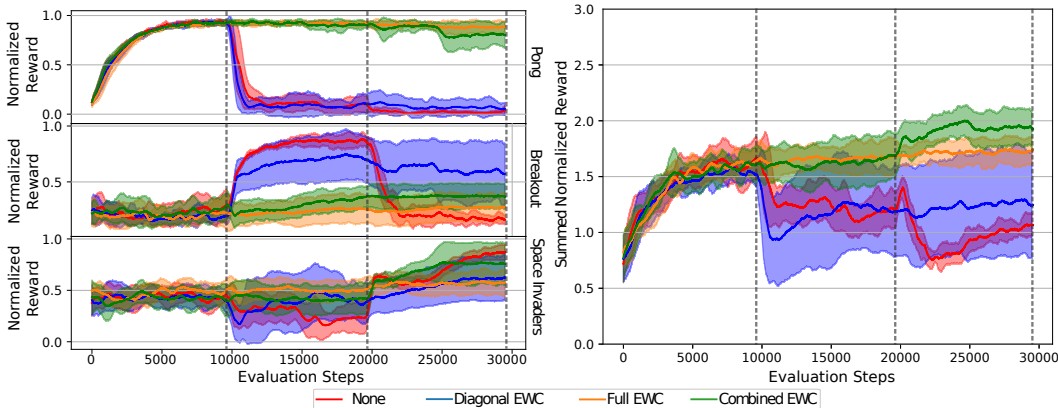

Figure 4: Results of the various EWC models trained to play three Atari domains (Pong, Breakout, Space Invaders) in order. (left) Mean standardized rewards in the three domains separately over the number of evaluation steps made by the agent. (right) Mean sum of the standardized rewards in the three domains over the number of evaluation steps from the agent. This demonstrates the aggregate performance of the model across all tasks. Once again we see a trade-off between the long-range memory of Full EWC and the ability to learn new tasks of Diagonal EWC. Importantly, Combined EWC can demonstrate both positive qualities to some degree - mitigating the trade-off.

over-parametrized. We cannot rule out that, if the network were significantly larger, the performance gap between Combined EWC and Diagonal or Full EWC would be removed. Similarly, based on our results it appears that Full EWC is better than Diagonal EWC when summing their rewards across the three environments. However, this hides the very different behaviour of the two regularizers which we found prioritize different information. Thus, it appears more appropriate to conclude that, in the not over-parametrized regime, these two regularizers are different strategies on a Pareto frontier of how the parameter space can be allocated. This is particularly clear when switching from Pong to Breakout which is difficult for all models and is likely due to the similar input and output spaces of the two environments. Something which prior theoretical and empirical results have identified as detrimental (Lee et al., 2022). To determine if this is indeed an adversarial case for EWC we provide results for an experiment run on the Boxman domain (Van Niekerk et al., 2019) where three tasks share an input and output space completely. Details are presented in Sec. A.7.2, however, the only difference between tasks in this domain is a colour icon at the top left of the input image which determines the colour of objects in the environment that the agent should aim to obtain. In this case we see no benefit from any version of EWC over a vanilla Double DQN. Thus, while our results demonstrate an important test of the various forms of EWC, the conclusions we are able to draw for ideal conditions such as having an over-parametrized network and distinct tasks are limited.

## 6    CONCLUSION

We see that there is some benefit towards using the full FIM in EWC as opposed to just the diagonal elements. However, using Full EWC alone consistently performs worse than Diagonal EWC as it is too conservative in restricting changes in parameter space. Consequently it stops new learning. By combining Diagonal and Full EWC we are able to obtain a regularizer which is flexible enough to allow for new learning and robust to forgetting, with the practical cost of optimizing two hyper-parameters instead of one. Most importantly, due to the Surrogate Hessian-Vector Product in Sec. 3, it is computationally very easy to add Full EWC to trainings and determine its benefit for new domains and tasks - which is the main contribution of this work. For example, the additional wall-clock time from adding Full EWC to the large NN in the RL experiment is $0.07558584$ seconds with a NVIDIA 1080 and jit compilation[1]. Thus, we have begun to answer the lingering open question with EWC of the performance costs associated with just using the diagonal of the FIM. We show that the loss in performance is not significant, however we see that there are performance gains from using the Combined EWC. We also begin to establish the conditions where the full FIM is most useful for EWC: wider and shallower networks initialized in the lazy regime of learning.

---

[1]All experiments are run using the Jax library (Bradbury et al., 2018)

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
