# OpenReview forum: "Full Elastic Weight Consolidation via the Surrogate Hessian-Vector Product"
_ICLR.cc/2024/Conference — ICLR 2024 Conference Withdrawn Submission_

### Official Review · Reviewer_HSfz · 2023-10-24

**Soundness:** 2 fair
**Presentation:** 2 fair
**Contribution:** 2 fair
**Rating:** 3
**Confidence:** 2

**Summary:**

This work studies the elastic weight consolidation (EWC) and tries to alleviate the computation burden of the Fisher Information Matrix (FIM). The main contribution is to propose a method for obtaining the full FIM. The performance is examined in the image classification and reinforcement learning.

**Strengths:**

This paper is relatively easy to follow but there still remains room to improve the organization. The vector product or matrix decomposition in learning is reasonable, and this trick most applied to sparsity, e.g. kernel matrix in Gaussian processes.
The examination of continual learning is relatively but not all standard in terms of benchmarks.

**Weaknesses:**

**1. About the layout of this paper.**

It seems there is too much background knowledge about the computation bottleneck in EWC from Page2 to Page4. There misses some parse in paragraph or key points are not well highlighted. These make it a bit difficult to follow in logics.

**2. About the contribution.**

The theoretical advantage to obtaining the full FIM is not well clarified in this work. Does it mean more information about the FIM brings better generalization in a theoretical sense? In the visual abstract, it seems the combined EWC can achieve more superiority than the full EWC, while this work focuses on the full EWC. Does this violate the research motivation? Meanwhile, it is necessary to connect some proposition to empirical observations.

**3. About the evaluation in reinforcement learning.**

I am afraid three Atari games are not typical in the continual learning domain. There exist more convincing benchmarks, e.g., Continual World, for continual reinforcement learning to examine the performance.

**Questions:**

See the weakness part.

---

> ### Author Response · Authors · 2023-11-15
> **Response to Official Review of Submission7245 by Reviewer HSfz**
>
> We thank the reviewer for their time. For point 3 we direct the reviewer to the General Comment. For point 2, we also direct the reviewer to the General Comment but also would like to clarify a couple points here. Firstly, this paper, and EWC in general, is not concerned with generalization. It aims to avoid catastrophic forgetting. Thus, our proposition shows that Full  EWC has a theoretical advantage in avoiding catastrophic forgetting in the lazy regime. Empirically we show that Combined EWC is better at avoiding catastrophic forgetting in general. Secondly, this work introduces both Full and Combined EWC and is focused on both. Thus, the visual abstract does not violate the research motivation, but summarizes our results. Combined EWC incorporates Full EWC and so fits within the remit of this work. The connection of the proposition and empirical observation is that the proposition explains the empirical observation that the relative benefit of using the Full FIM is greater in the lazy regime. We direct the reviewer to the first paragraph of Section 4 where this point is made. If the reviewer has suggestions on how to improve on this paragraph we will gladly incorporate them.
>
> Finally, we thank the reviewer in highlighting the usefulness of vector products and matrix decompositions in machine learning algorithms as well as the connection to sparsity in Gaussian process kernels. However, we kindly note that our SHVP is directly equivalent to the Hessian and not a sparse approximation.
>
> We hope that this addresses all of the reviewers' concerns and are happy to engage further during this discussion period.

---

> > ### Author Response · Authors · 2023-11-20
> > **Following Up with Reviewer HSfz**
> >
> > We would once again like to thank the reviewer for their time in reviewing our paper. We would also like to ask if we have addressed all of the reviewer's concerns and see if there is anything further that we can clarify as the discussion period draws to a close.

---

### Official Review · Reviewer_mrqD · 2023-10-31

**Soundness:** 3 good
**Presentation:** 3 good
**Contribution:** 2 fair
**Rating:** 5
**Confidence:** 3

**Summary:**

The paper introduces the SHVP (Surrogate Hessian Vector Product) algorithm, as a means to compute the product of the Hessian matrix of a neural network with a given vector, without needing to compute the matrix itself (thus significantly reducing the computational complexity), but by relying on two nested backpropagations instead.

Using this, the authors are able to train models in a continual learning context, using the Elastic Weight Consolidation (Kirkpatrick et al. 2017) training objective to avoid catastrophic forgetting, while using the complete Hessian matrix, as opposed to a diagonal approximation as is often used in practice for computational reasons.

The paper experimentally shows that using the full Hessian matrix improves the model's capacity to retain its performance on older tasks, and that the best performance is obtained by combining the full EWC with its diagonal approximation, allowing an even equilibrium between older and more recent tasks.

**Strengths:**

The article is clear, and provides good background and context to position its work in the larger literature.

The proposed algorithm is sound and well justified, and the experimental exploration of the impact of the different variants of EWC, in relation with the two training regimes is an interesting contribution.

Overall, I believe the SHVP algorithm introduced here can be an useful tool to explore the Hessian matrices of neural networks.

**Weaknesses:**

I have one main issue with the paper and its contribution: the positioning of the authors with regard to how and when their proposed Combined EWC could be used is very unclear.

As far as I can tell, the proposed SHVP algorithm trades the one-time computation of the full Hessian of the model for a double backprop through the NN at every iteration of the training. In particular, this means that computing the gradient associated with the EWC regularization term *requires computing an expectation over the datasets of the previous tasks*.

My understanding is that the main appeal of EWC is to keep some kind of summary of the previous tasks as the Hessian of their losses (approximated as diagonal or block-diagonal for computational efficiency). This relies on the idea that keeping around the training data of the previous tasks is either not desirable or not possible, otherwise one would simply train their model on all tasks simultaneously.

Here, the proposed "Combined EWC" requires keeping access to the data of the previous tasks, and appears to be more computationally expensive than just training the model on the multiple tasks simultaneously.

As a result, it is unclear to me what does "Combined EWC" actually bring to the table: while not stating it clearly, the paper frames it as an algorithm that could be used in practice to train models in the continual learning setting. But given the above remarks, I fail to see when one would actually want to do that.

**Questions:**

**How is the computation defined by Proposition 1 done in practice?**

The paper and appendix are not very detailed about it, and the joined code is barely documented, making it difficult to understand. I don't see how it would be possible to compute the double gradient defined for SHVP without retaining the whole computational graph associated with task A in memory (for performing the second backprop).

**When would someone use SHVP or Combined EWC in practice?**

Given the previous discussion, I'm having a difficult time figuring when one would opt for using Combined EWC, instead of simply training the model on multiple tasks simultaneously, given EWC is supposed to be an approximation of that. What point is there for me to use an approximation that is not cheaper to compute than the actual thing?

---

> ### Author Response · Authors · 2023-11-15
> **Response to Official Review of Submission7245 by Reviewer mrqD**
>
> We thank the reviewer for their time and praise of many aspects of our work. We are glad to see that the reviewer also acknowledges the connection to the training regimes as being a contribution.
>
> The reviewer’s two main points appear to be summarized in the questions and so we will respond to those. For Question 2 on when to use Combined EWC we direct the reviewer’s attention to the General Comment. However, to very briefly summarize, we provide a way to make Combined EWC nearly as efficient as Diagonal EWC. Thus, we would recommend using Combined EWC whenever EWC is an appropriate algorithm to use in practice. Diagonal, Full and Combined EWC should all be used in settings where continual learning is necessary. So as the reviewer correctly points out, data for Task A is lost after moving on to Task B for example.
>
> For Question 1 on how the computation is done and its efficiency, we are glad to say that we have a clear answer for this. The first backward pass of the gradient at the end of a previous task needs to only be computed once. So to add the Full EWC regularization term when switching to a second Task B, at the end of Task A you store $\theta_A$ (the optimal parameters for the previous task, as is always done for EWC) and the first derivative vector at the optimal point for Task A ($\nabla_{\theta_A}L(\theta_A)$). By computing and storing this gradient vector you do not need the dataset for Task A anymore and this is only computed once. To then update the parameters for Task B **we do not use two nested backpropagations**. Thus, the point which says “the proposed SHVP algorithm trades the one-time computation of the full Hessian of the model for a double backprop through the NN at every iteration of the training.” is not true. Backprop is used once an iteration. This is **the main contribution of our work**. To actionably address the reviewer’s concerns we will add pseudocode to the appendix making this clear, and we also direct the reviewer’s attention to Figure 1 and the paragraph directly next to it.
>
> We hope that this addresses all of the reviewers' concerns and are happy to engage further during this discussion period.

---

> > ### Comment · Reviewer_mrqD · 2023-11-20
> >
> > I'm sorry, but this answer does not ease my concern, quite the opposite actually.
> >
> > You construct the EWC gradient as follows:
> > $$
> > \\nabla_\\theta EWC(\\theta) = \\nabla_{\\theta_A} \\Big[ (\\theta - \\theta_{A_{const}})^T \\; \\nabla_{\\theta_A} L_A(\\theta_A) \\Big]
> > $$
> >
> > If, as you claim, $\nabla_A L_A(\theta_A)$ is computed once and for all, and no double backprop is performed, then in the training for later tasks, it behaves like a constant wrt gradient computation. As a result, in the above formula, none of the terms have a gradient wrt $\theta_A$, and this would mean that $\nabla_\theta EWC(\theta)$ would be effectively computed as identically zero, rendering your proposed algorithm meaningless.
> >
> > So as far as I can tell, given your code does apparently not produce such an identically zero gradient, either you are actually performing double-backprop without realizing it, or your code is not actually computing the same thing as the text of your paper claims it is. In both cases it is in my opinion a strong issue with your submission.

---

> > > ### Author Response · Authors · 2023-11-20
> > > **Response to Reviewer mrqD**
> > >
> > > We thank the reviewer for engaging in this discussion period and giving us the opportunity to clarify our methodology.
> > >
> > > For the function which updates the network parameters we pass in four arguments. The current network parameters $\theta$. The network parameters at then end of the previous task - set to be constant $\theta_{A_{const}}$. The first derivative at the end of the previous task $\nabla_{\theta_A}L_A(\theta_A)$ and a batch of data for the current task $(X,Y)$. We calculate the loss as normal with $E[L(\theta, (X,Y))]$ and compute the gradient with $\nabla_\theta E[L(\theta, (X,Y))]$. Similarly, we need to calculate the derivative of the EWC term $\nabla_\theta EWC$ which used $\theta$, $\theta_{A_{const}}$ and $\nabla_{\theta_A}L_A(\theta_A)$. As far as we can tell the reviewer recognizes that this computation of $\nabla_\theta EWC$ does not actually make a call to $\nabla_\theta$ but instead uses a call to $\nabla_{\theta_A}$ when computing $\nabla_{\theta_A}[(\theta - \theta_{A_{const}})^T \nabla_{\theta_A}L_A(\theta_A)]$. In reality all we have done is applied the sum rule manually and then manipulated the EWC (regularization) term.
> > >
> > > Are we correct in saying that the reviewer is concerned that two different gradients are being called - one for each term: $\nabla_\theta$ for the loss and $\nabla_{\theta_A}$ for the EWC term?
> > >
> > > We will assume that this is the reviewer's intended meaning and ask that we be corrected if we miss the point. If in the original review the phrase "double backprop" was intended to mean two separate calls to the gradient computation ($\nabla_\theta$ and $\nabla_{\theta_A}$) separately. Then this is correct (and not how we had interpreted it). Note that because these gradient computations happen separately this has $\mathcal{O}(N)$ complexity. In the context of a Hessian calculation "double backprop" could also be interpreted as a gradient computation within a gradient computation (in practice this is usually made slightly better by using backward and then forward mode differentiation in succession) with a complexity of $\mathcal{O}(N^2)$. This was the interpretation we had from the initial review, also because the reviewer was asking whether an expectation over the previous dataset was needed at each timestep (which would indicate the Hessian is being recalculated). So to be clear here. The second interpretation of the two nested gradient calls is not true. The first interpretation of two distinct gradient calls is true. But this is still in line with all claims of efficiency made in this work of Full EWC with the SHVP being $\mathcal{O}(N)$ memory and computation complexity. Additionally, this is no worse than calculating the gradient (with a single gradient call) for any update rule with both a loss and regularization term: $\nabla_\theta [L(\theta, (X,Y)) + R(\theta)]$. This is because the sum rule would compute these derivatives separately anyway as $\nabla_\theta L(\theta, (X,Y)) + \nabla_\theta R(\theta)$. So really our computation of $\nabla_\theta L(\theta, (X,Y)) + \nabla_{\theta_A}[(\theta - \theta_{A_{const}})^T \nabla_{\theta_A}L_A(\theta_A)]$ is not different in complexity to any update rule with a regularization term.
> > >
> > > We note that Appendix A.1 provides empirical evidence for the accuracy of the SHVP (in a far more challenging setting than being used to calculate the EWC term). The Visual Abstract (b) shows the computational efficiency of Full EWC with the SHVP compared with the full Hessian (on a small setting where computing the full Hessian is feasible to begin with). Finally, it is just not possible to hold the full Hessian for such large NNs as we use in our experiments. We just could not perform a computation of $\mathcal{O}(N^2)$ memory complexity, and so this supports the memory complexity of our proposed method being better than computing the full Hessian.
> > >
> > > We reiterate that we are thankful to the reviewer for their meticulous review. We will use these questions to guide the edits we make to the revised paper. Finally, we are glad to continue this discussion if more issues persist over this discussion period.

---

> > > > ### Comment · Reviewer_mrqD · 2023-11-20
> > > >
> > > > I'm sorry, but it seems to me you are not understanding my point. Your initial understanding of « double backprop » was correct, and your followup answer still does not answer my concern, which is a mathematical one. To phrase it in another way, my point is that **Proposition 1 does not hold if you are not doing double-backprop thought $\\nabla_ {\\theta_A} L_A(\\theta_A)$.**
> > > >
> > > > Allow me to be more explicit about the mathematical issue:
> > > >
> > > > Let us name $g_A = \\nabla_A L_A(\\theta_A)$. According to your reasoning, this term is computed once and for all at the end of task A, and the computational graph is not retained. Thus, as far as the backprop algorithm is concerned, this is a constant.
> > > >
> > > > Now, what you are computing is thus the gradient wrt to $\\theta_A$ of the quantity $ (\\theta - \\theta_{A_{const}})^T g_A$, but this expression is a constant wrt $\\theta_A$:
> > > > - $\\theta$ and $\\theta_A$ are different variables, so $\\nabla_{\\theta_A} \\theta = 0$
> > > > - $\\theta_{A_{const}}$ is (rightly) treated as a constant as well, so $\\nabla_{\\theta_A} \\theta_{A_{const}} = 0$
> > > > - You claim to not proceed to a double-backprop, so $g_A$ is also a constant: $\\nabla_{\\theta_A} g_A = 0$
> > > >
> > > > As a result, if your code actually does what you are claiming it is doing, then the gradient you are computing should be identically zero.
> > > >
> > > > I thank you for taking the time to answer this issue, because in my opinion it is a serious one. Indeed in my initial review, I assumed your algorithm was doing double-backprop even if you don't say it explicitly, because it is what is required for your algorithm to be mathematically valid.
> > > >
> > > > Looking again at your code, and from what I understand of how `jax` works, it seems to me that you are not actually computing the gradient of $ (\\theta - \\theta_{A_{const}})^T \\nabla_{\\theta_A}L_A(\\theta_A)$ with regards to $\theta_A$, but with regards to $\\nabla_{\\theta_A}L_A(\\theta_A)$. If that is indeed the case, that means that what your code is actually computing is $\\nabla_\\theta EWC(\\theta) = (\\theta - \\theta_{A_{const}})$, which would be equivalent to using the regularizer $EWC(\\theta) = \\frac{1}{2}\\|\\theta - \\theta_A\\|^2$.

---

> > > > > ### Author Response · Authors · 2023-11-21
> > > > > **Response to Reviewer mrqD**
> > > > >
> > > > > We thank the reviewer for their meticulous evaluation of our work, including by checking our code.
> > > > >
> > > > > We see the reviewer's point and acknowledge that they are correct: Jax is not tracking the gradients sufficiently for our implementation of Full EWC to be working as indented. Consequently, the experiments presented do not match the mathematics or ideas expressed and are inadmissible. Regrettably, we will be retracting our paper.
> > > > >
> > > > > We are grateful to the reviewer for catching this implementation error in our code and their patient discussion with us.

---

### Official Review · Reviewer_gTVN · 2023-10-31

**Soundness:** 2 fair
**Presentation:** 2 fair
**Contribution:** 2 fair
**Rating:** 3
**Confidence:** 5

**Summary:**

This paper introduces a method to calculate the full Fisher Information Matrix for EWC (instead of the more common diagonal Fisher) that is computationally and memory efficient. The paper then argues that combining the full EWC with a diagonal EWC is better than just one or the other. The paper then argues that EWC works better in the lazy training regime (with large parameter initialisations). Finally, in Section 5, the paper applies their method to an RL problem (3 Atari games sequentially shown).

**Strengths:**

1. I really like the algorithm for Full EWC in Section 3. I like that the authors looked at empirically verifying it in Appendix A.1, and would have liked to see more of this!

2. Applying to RL / Atari is not done often enough in continual learning, so it was nice to see that experiment in this paper.

3. I liked the proof sketches in the main text; I thought they were well-written and useful.

**Weaknesses:**

1. It is important to talk about other related works that use eg block-diagonal Fishers for EWC in Section 2. For example, Ritter et al., 2018 (A Scalable Laplace Approximation for Neural Networks).

2. Permuted MNIST is an old benchmark with many problems. It is difficult to draw many conclusions of note from it due to the various issues with it (see for example Farquhar and Gal (Towards Robust Evaluations of Continual Learning) or Swaroop et al (Improving and Understanding Variational Continual Learning) for discussions). Additionally, the authors in this paper only use 5 tasks, which is very few, making it difficult to draw any real conclusions. It would be great if the authors ran for significantly more tasks to see if their conclusions/results still hold (eg 20 tasks).

3. When comparing Figures 2 and 3, I'm not really sure if the lazy training regime is better for EWC! It looks like Figure 2 often has higher performance, at least on everything but task A's accuracy after training on the last task. Could the authors report other metrics like overall average accuracy (and maybe forward/backward transfer) too?

4. Proposition 2 seems like it is re-writing that, under conditions like constant covariance (ie a quadratic loss landscape?) and mean-squared error loss, that Laplace approximation is ideal. I am not sure that there is anything new here: this is the reason that eg the EWC paper used the Laplace approximation / FIM in the regulariser / Bayesian approach. I found it odd that, in the Appendix, the authors prove this via "Bayes Optimal estimators" (ie looking at predictions at a test point?) and then take the limit as the Gaussian approximation becomes a delta (ie reduce the covariance to 0). This looks like a simple MAP estimation to me? Please let me know if I am missing something here.

5. In the text on page 7, the authors argue why the landscape may be (more) locally convex in the lazy training regime, because parameter values do not change much during training. I am not sure I am convinced by this: just because the parameter values do not change (relatively) very much, this does not mean that the loss landscape that the parameters move through is better-behaved: it could still be highly non-convex. Is there previous literature on this (or could the authors design an experiment to show this)?

6. I do not understand the intuition for why Combined EWC is better than Full EWC or diagonal EWC. Diagonal EWC seems to perform very well in Figures 2 and 3 already. In Figure 4, it does not forget task 2 when training on task 3 (although it does forget task 1), against the authors' conclusion (that Diagonal EWC prioritises new tasks). I think I need much more evidence of these claims (eg that Diagonal EWC prioritises new tasks while Full EWC prioritises old tasks) to believe them sufficiently.
- Also, although it is nice to have an RL experiment in Section 5, having only 3 tasks/games is too few to draw conclusions.

**Questions:**

Please see Weaknesses section. More minor questions / comments:
1. Note that the authors are using the empirical Fisher, not just the Fisher / FIM, in Equation 1 (and throughout the paper). See for example Kunstner et al., 2020 (Limitations of the Empirical Fisher Approximation for Natural Gradient Descent) for a discussion.
2. I think, in the text after Equation 6, the authors meant 'low uncertainty' and not 'high uncertainty'?
3. At the bottom of page 8 (and in Sec A.7.1) the authors use a network of half the size and see lower performance. I do not see how this helps understand if the experiment in Section 5 is sufficiently over-parameterised or not.

---

> ### Author Response · Authors · 2023-11-15
> **Response to Official Review of Submission7245 by Reviewer gTVN (Part 1 of 2)**
>
> We thank the reviewer for their time, constructive comments and kind acknowledgement of a number of our work’s strengths. We will address each of the points raised as Weaknesses below and then subsequently answer the points raised as Questions after.
>
> Weaknesses:
> 1. On the discussion of Ritter et. al. (2018) [1]. We will gladly add this citation. We would like to point out that we do cite McInerney et. al. (2021) [2] which is a more recent work building directly on the work of Ritter et. al. (2018) and we do mention this work in our Introduction.
> 2. Please see the general comment for the overall discussions on our experiments. We would just like to elaborate on one point here. This experiment uses dense neural networks. Thus, arguments around the unnatural nature of the permuted images [3] are not as relevant in a network without architectural biases such as convolution which assume spatial contiguity of objects. Additionally, avoiding catastrophic forgetting with a dense network is difficult (as our experiments highlight), even on such a simple task with an ideal input space. Consequently, even after five tasks we see a clear result on the behaviour of each of the algorithms. Moreover, [4] makes the point that Permuted MNIST is not appropriate for comparison when network architecture or other model design decisions change. However, in our controlled experiments all models and benchmarks use the exact same architecture. We believe that these points, coupled with the general comment above, justify the use of Permuted MNIST in our case. We will add the two given citations to our revised version and use them to be clearer about the design decision and limitations of our experiments - as we already do with work making a similar point such as [5].
> 3. The reviewer is indeed correct, the feature learning regime does out-perform the lazy regime. However, this is not the point we were making in Section 4. In this section we note that the relative benefit of using the full FIM in the lazy regime is better than the feature learning regime. In other words, the comparison is not between the green lines in Figure 2 and Figure 3, but rather a comparison of how much higher the green line is over the blue and orange in Figure 3, compared to the same gap in Figure 2. It is a more subtle point, and we will aim to make this clearer in the revised version of the paper, by more clearly annotating Figure 2 and 3. We will also elaborate further in both figures' captions. We thank the reviewer for drawing this to our attention. We hope it is clearer now that what matters is that the green curve is much higher than the blue and orange curves in Figure 3, but only slightly higher in Figure 2 (note it still gets the best of both blue and orange curves in both cases).
> 4. and 5. We believe points 4. and 5. go together and so we address them at once. We  hope that the clarification on point 3 will also help here. Proposition 2 is indeed similar to a MAP proof of optimality while also connecting the covariance of the parameter space to the FIM. This is the basis of the original EWC work. Our contribution here is to connect the assumptions necessary for this optimality to hold, to the two regimes of training which have been identified in the literature [6]. To our knowledge this is a new connection and one which we found due to the results of Permuted MNIST - that the **relative performance** when including the full FIM is greater in the lazy rather than feature learning regime. This leads to our response for point 5. In the lazy regime the parameters, which are initialized with a Gaussian of fixed covariance, do not change from their initial values much. Thus, they remain distributed by the same Gaussian from which they are sampled. As a result the necessary assumptions for the MAP proof to hold are true. In the feature learning regime, parameters can change and become potentially more coupled [7]. So the same argument (and the assumptions for EWC to be optimal) do not hold. Finally, it is indeed well established in the literature that in the lazy regime (often called the NTK regime [6]) the local loss landscape is smooth and in the infinite width limit (with the actual NTK) [8] the loss landscape is exactly convex [9,10,11]. We mention this in the last paragraph of page 7 but will gladly provide more citations as the reviewer requests, since this is a well-established point [6,7,8,9,10,11].

---

> > ### Author Response · Authors · 2023-11-15
> > **Response to Official Review of Submission7245 by Reviewer gTVN (Part 2 of 2)**
> >
> > 6. We suspect there is a potential terminological issue, and if so, we are happy to revise our terminology for clarity. Our point is that Diagonal EWC is better at retaining knowledge of recent tasks compared to Full EWC. We don't aim to imply that Diagonal EWC is prone to forgetting all prior tasks when learning new ones. For example, during training on task $N$ Diagonal EWC tends to retain knowledge of task $N-1$, while Full EWC leans towards preserving information from task $1$ (this example is taken somewhat to extremes for argument sake). Our claim doesn't revolve around the algorithms' ability to learn task $N$. Instead it is about the representations they acquire and preserve. Full EWC has a rigid representation, whereas Diagonal EWC allows more flexible representations with a tendency to drift. This drift allows the retention of the most recently influenced task in the representation but not longer-term memory. Therefore, the reviewer’s comment on Figure 4 aligns with our observations - Diagonal EWC prioritizes remembering the recent task, while Full EWC prioritizes earlier tasks. These conclusions for Figure 4 have also been influenced by our observations in Figures 2 and 3. Despite the value of additional evidence, we believe our experiments are valid and useful (please also refer to the General Comment above) and are open to alternative interpretations. Our final point is that Combined EWC offers a more precise control over representations, allowing certain parameters to drift while maintaining rigidity in others. Consequently, Combined EWC either matches or surpasses Full EWC and Diagonal EWC in all our experiments.
> > 7. On the point of needing more RL experiments we once again refer to the General Comment and Appendix A.7 in our paper.
> >
> > Questions:
> > 1. The limitations of the Empirical FIM will be true of any work on EWC. Unfortunately, to our knowledge, there is presently no way around this.
> > 2. Yes, thank you for catching that. We will correct this in the next revision.
> > 3. If the network were over-parametrized then you could reduce its size and see no drop in performance. Here we reduce its size and see a drop in performance. Thus, it would seem the original network is not significantly over-parametrized (at least not to the point that it can be halved). This is however, a small supplementary experiment and so we do not lean heavily on these results as dropping the network size in half is quite a big drop, and so it is difficult to say anything too precise.
> >
> > We hope that this addresses all of the reviewers' concerns and are happy to engage further during this discussion period. Additionally, we hope that we have clarified the intention and academic context in which we place this work.
> >
> > [1] Ritter, Hippolyt, Aleksandar Botev, and David Barber. "A scalable laplace approximation for neural networks." 6th International Conference on Learning Representations, ICLR 2018-Conference Track Proceedings. Vol. 6. International Conference on Representation Learning, 2018.\
> > [2] McInerney, Denis Jered, et al. "Kronecker factorization for preventing catastrophic forgetting in large-scale medical entity linking." arXiv preprint arXiv:2111.06012 (2021).\
> > [3] Farquhar, Sebastian, and Yarin Gal. "Towards robust evaluations of continual learning." arXiv preprint arXiv:1805.09733 (2018).\
> > [4] Swaroop, Siddharth, et al. "Improving and understanding variational continual learning." arXiv preprint arXiv:1905.02099 (2019).\
> > [5] Lee, Sebastian, et al. "Maslow’s Hammer in Catastrophic Forgetting: Node Re-Use vs. Node Activation." International Conference on Machine Learning. PMLR, 2022.\
> > [6] Geiger, Mario, et al. "Disentangling feature and lazy training in deep neural networks." Journal of Statistical Mechanics: Theory and Experiment 2020.11 (2020): 113301.\
> > [7] Braun, Lukas, et al. "Exact learning dynamics of deep linear networks with prior knowledge." Advances in Neural Information Processing Systems 35 (2022): 6615-6629.\
> > [8] Jacot, Arthur, Franck Gabriel, and Clément Hongler. "Neural tangent kernel: Convergence and generalization in neural networks." Advances in neural information processing systems 31 (2018).\
> > [9] Lee, Jaehoon, et al. "Wide neural networks of any depth evolve as linear models under gradient descent." Advances in neural information processing systems 32 (2019).\
> > [10] Lee, Jaehoon, et al. "Deep Neural Networks as Gaussian Processes." International Conference on Learning Representations. 2018.\
> > [11] Pleiss, Geoff, and John P. Cunningham. "The limitations of large width in neural networks: A deep Gaussian process perspective." Advances in Neural Information Processing Systems 34 (2021): 3349-3363.

---

> > ### Comment · Reviewer_gTVN · 2023-11-16
> > **Response**
> >
> > Thanks to the authors for their rebuttal.
> >
> > 1. Thank you for pointing this out to me.
> >
> > 2. My issue is more that there is not enough evidence if one just looks at Permuted MNIST. There are a host of things going on with Permuted MNIST (such as (i) 5 tasks being far too few to test any continual learning algorithm given the model size, and (ii) each new task is so different, in terms of gradients, to others, which is usually very unrealistic), and I need to see more evidence on other benchmarks before I believe the claims being made.
> >
> > 3. Again, thanks for pointing this out. However, I really am not sure I agree with the authors' conclusions. Comparing the difference between green and blue lines in Figure 2 vs 3: the difference looks similar to me! But I can only inspect this visually. Perhaps the authors could devise some metric to argue this point? Comparing the green and orange lines: I can see that there is maybe a difference (mainly in the top left panel?). However, Orange is 'Full EWC', and it now seems like Full EWC performs worse (compared to diagonal) in the lazy regime (ie in Figure 3)?
> >
> > 4. and 5. In my mind this is not a surprising or interesting result. In the NTK regime, where covariances remain similar, full EWC is always going to do very well, and this is a straightforward application of Bayes rule (and the motivation for introducing EWC as an algorithm in the first place). I realise this is my opinion, and others may disagree that this is obvious. So perhaps the authors, other reviewers and/or the AC can disagree with me, and that can go into the decision-making for this paper.
> >
> > 6. I see, I clearly misunderstood earlier. Similarly to in point 3, I recommend the authors try to formalise this more. For example, by having some simple metric that measures performance drop for just the last task vs performance drop in previous tasks. This would help sell this much better. I would then also repeat the experiment on more benchmarks and confirm that these trends hold!
> >
> > 7. I realise how annoying it is to be asked for more experiments, but I do really think that in this kind of paper -- where you are (i) introducing a new algorithm that you say is computationally efficient, and (ii) making empirical claims regarding Combined EWC -- it is important to show trends (particularly for (ii)) hold. I really think this would make the paper much stronger.
> >
> > A final passing comment (as potential future work!): another way that this work could be significantly improved would be to look theoretically into why Diagonal EWC and Full EWC might be acting in the ways they do (in terms of performance on the previous task vs tasks a long time ago). Some theory here would be extremely interesting, I think.

---

> > > ### Author Response · Authors · 2023-11-17
> > > **Response to Reviewer gTVN (Part 1 of 2)**
> > >
> > > We are glad to see that we have clarified some important points. We thank the reviewer for engaging in this discussion period and their consideration of our rebuttal.
> > >
> > > It seems we have converged on two main discussion points. That of the NTK regime result and the need for further or other experimentation.
> > >
> > > On the point of the experiments. We will start with the two points made on Permuted MNIST. On the point (i) which says “5 tasks being far too few to test any continual learning algorithm given the model size” - clearly not in our experiments as Diagonal EWC sees a performance drop of 20% on Task A over the 5 tasks. On point (ii) which says “each new task is so different, in terms of gradients, to others, which is usually very unrealistic” - this is at least not how the lazy regime works as it is not doing feature learning, and instead just learns the readout weights which all connect to the same output space and use the same final hidden layer latent space [9]. But even then, in the feature learning setting, if this were true Diagonal EWC accuracy would not drop by 20% over the 5 tasks. Neural networks do not implicitly disentangle their hidden neurons [12], nor do they promote sparsity in all cases [13]. Some cases have been found where, with a particular activation function, learning rule and dataset, portions of the hidden layer specialize [14,15]. However, even with such specialization it has been shown that subsequent tasks will re-use these specialized latent representations and not the unoccupied portion of the hidden layer [5]. Thus, it certainly seems possible that hidden representations and gradients could interfere between tasks. The excess capacity of the network matters primarily if we know the network is leaving portions of the latent space completely unused for subsequent tasks. Is there a citation which speaks to the gradients or latent sparsity of the network in Permuted MNIST which we are unaware of?
> > >
> > > We also note that three main problem paradigms have been identified in the literature for continual learning [16]. These are:
> > > 1. Task-Incremental Learning: a model must solve multiple distinct tasks.
> > > 2. Class-Incremental Learning: a model must solve a classification problem, where each new task adds a class.
> > > 3. Domain-Incremental Learning: a model must perform the same task, but the input features change between tasks.
> > >
> > > Clearly Permuted MNIST is an appropriate task for evaluating Domain-Incremental Learning and thus is appropriate to evaluate this paradigm of continual learning. We will be glad to also add this distinction between Task Incremental and Domain Incremental Learning to our work too. We note that the primary problem with Permuted MNIST presented in [3] is: “*Cross-task resemblances: Input data from later tasks must resemble old tasks enough that they at least sometimes result in confident predictions of old classes, early in training. The widely used Permuted MNIST which violates this would correspond to every input sensor in our Mars rover being randomly rewired—unlikely to be representative of real cases*”. However the work which originally introduces Permuted MNIST [17] states: “*To test this kind of learning problem, we designed a simple pair of tasks, where the tasks are the same, but with different ways of formatting the input. Specifically, we used MNIST classification, but with a different permutation of the pixels for the old task and the new task. Both tasks thus benefit from having concepts like penstroke detectors, or the concept of penstrokes being combined to form digits. However, the meaning of any individual pixel is different. The net must learn to associate new collections of pixels to penstrokes, without significantly disrupting the old higher level concepts, or erasing the old connections between pixels and penstrokes*”. Thus, there is still room for disruption of high level concepts which EWC must work to avoid, especially in a network which does not have implicit or enforced sparsity. We do not dispute with [3] that Permuted MNIST is somewhat idealic, our argument is that it is able to still provide a useful experiment. We advocate for a degree of nuance beyond a complete rebuke of what has been [18] and continues to be a widely used and interpretable domain [19]. As [19] (which was presented at NeurIPS after [3]) note, there is still some benefit to Permuted MNIST. They go on to perform two, more sophisticated supervised learning tasks to monitor forgetting, while we perform two, more sophisticated RL experiments (within the realm of what EWC can learn).

---

> > > > ### Author Response · Authors · 2023-11-17
> > > > **Response to Reviewer gTVN (Part 2 of 2)**
> > > >
> > > > Similarly for point 6(i). Storing the gradient vector between tasks is significantly less computationally intensive than storing the Hessian. This doesn’t require experimental results or a trend. It is analytical and certain. But on top of this, Visual Abstract (b) clearly shows extremely different scalings between the two. On point (ii) we agree that more experiments would make the paper stronger. What matters is whether the experiments we present are controlled and provide results of sufficient interest to the field. We believe so.
> > > >
> > > > On the point of the NTK. We do **not** agree that this result is not interesting. Further, now that we have laid out the argument of the NTK regime and origins of EWC in Bayes rules, the result may appear unsurprising. But in many cases, once the argument is made, the conclusion does appear clear. We have shed light on a link between two subfields, one which had previously not been noted. We would like to note that Reviewer mrqD even mentions “The proposed algorithm is sound and well justified, and the experimental exploration of the impact of the different variants of EWC, in relation with the two training regimes is an interesting contribution”.
> > > >
> > > > It is true that EWC is based on Bayes rule where the intuition is that previous tasks act as priors on the parameter distribution for the current task. To our knowledge **the Bayes optimality of full EWC, however, has never been shown**. The Bayes optimality of a linear model with Gaussian prior and likelihood (quadratic loss) is known. But this result has been missing from EWC because the connection to the lazy regime was needed such that the assumptions for a conjugate Gaussian prior optimality proof became valid. We reiterate, the contribution here is that we connect EWC to the lazy regime of training where the standard conjugate Gaussian prior proof is valid. But rephrasing a difficult problem in a way which makes it simple but connects it to an entirely different subfield is not uninteresting - it is one of the most useful tools in the mathematical toolkit. This is also why we call our formal statement a proposition - it follows directly from a theorem (that of the Bayes optimality of the MAP estimator).
> > > >
> > > > Finally, we appreciate the reviewer’s suggestion of future work, and take it as a sign of the potential impact this paper may have. But the reviewer’s suggestion to “look theoretically into why Diagonal EWC and Full EWC might be acting in the ways they do” and “Some theory here would be extremely interesting” is exactly what Section 4 aims to do. More work is needed, but theory tends to be slow and hard fought. Again, we believe the theory we do contribute here is meaningful and look forward to the future work that builds on it. Surely the new connection we make with the NTK regime will provide a clear next step in the development of the theory.
> > > >
> > > > [3] Farquhar, Sebastian, and Yarin Gal. "Towards robust evaluations of continual learning." arXiv preprint arXiv:1805.09733 (2018).\
> > > > [5] Lee, Sebastian, et al. "Maslow’s Hammer in Catastrophic Forgetting: Node Re-Use vs. Node Activation." International Conference on Machine Learning. PMLR, 2022.\
> > > > [9] Lee, Jaehoon, et al. "Wide neural networks of any depth evolve as linear models under gradient descent." Advances in neural information processing systems 32 (2019).\
> > > > [12] Locatello, Francesco, et al. "Challenging common assumptions in the unsupervised learning of disentangled representations." international conference on machine learning. PMLR, 2019.\
> > > > [13] Dasgupta, Ishita, Erin Grant, and Tom Griffiths. "Distinguishing rule and exemplar-based generalization in learning systems." International Conference on Machine Learning. PMLR, 2022.\
> > > > [14] Goldt, Sebastian, et al. "Dynamics of stochastic gradient descent for two-layer neural networks in the teacher-student setup." Advances in neural information processing systems 32 (2019).\
> > > > [15] Whittington, James CR, et al. "Disentanglement with biological constraints: A theory of functional cell types." The Eleventh International Conference on Learning Representations. 2022.\
> > > > [16] Van de Ven, Gido M., and Andreas S. Tolias. "Three scenarios for continual learning." arXiv preprint arXiv:1904.07734 (2019).\
> > > > [17] Goodfellow, Ian J., et al. "An empirical investigation of catastrophic forgetting in gradient-based neural networks." arXiv preprint arXiv:1312.6211 (2013).\
> > > > [18] Kemker, Ronald, et al. "Measuring catastrophic forgetting in neural networks." Proceedings of the AAAI conference on artificial intelligence. Vol. 32. No. 1. 2018.\
> > > > [19] Mirzadeh, Seyed Iman, et al. "Understanding the role of training regimes in continual learning." Advances in Neural Information Processing Systems 33 (2020): 7308-7320.

---

### Author Response · Authors · 2023-11-15
**General Comment to Reviewers**

We once again thank the reviewers for their time. There is a general sentiment among the three reviews regarding the general use of EWC in continual learning. We would like to discuss this point as a general comment and hope this might promote clear engagement on this point.

Firstly, we acknowledge that the use of EWC for continual learning as of the time of writing - be it the original Diagonal EWC [1] or Full and Combined EWC as proposed in this work - is limited. For example, we aimed to include an experiment on RL in this work as we believed it would be a significant weakness to omit this. However, we are unaware of any work which has improved EWC’s performance in RL beyond Atari games without also employing more elaborate techniques [2]. Thus, we use three Atari games in this experiment as they are simple enough for EWC to learn these tasks, avoid **some** catastrophic forgetting and present interpretable results. We are aware that these domains are not standard in modern continual learning research (although they were used in the original EWC paper [1] and mentioned recent work [2]). Thus, we present a more realistic continual learning RL experiment in Appendix A.7 with the Boxman environment [3]. However, all algorithms struggle on this task (Combined EWC least of all though). Thus, this work in no way proposes Full or Combined EWC as state of the art. So the first point here is to say, the fact that we use EWC in isolation limits the set of concise and controlled experiments we can run.

This leads then to a question of our contribution, if we are not proposing a new SOTA model, when should Combined EWC be used and what is the point of this work? Firstly, we recommend Combined EWC be used **whenever** EWC is an appropriate model for the given task. We have provided an extremely computationally efficient way to switch from Diagonal to Combined EWC. We have demonstrated that this promotes continual learning over a longer sequence of tasks compared to Diagonal EWC. Finally, we have shown that this is particularly useful in the lazy training regime. However, we do not necessarily recommend EWC methods in all cases. Hence, we have aimed to be very clear about the limitations in this work. The last paragraph of page 8 and all of the written components of page 9 discuss the limitations of EWC and our experiments. Thus, for much of what the reviewers raise as concerns, we agree with and already acknowledge in this limitations discussion. We do think, however, that this work still presents useful experiments, a computational tool and connection to loss landscape geometry which is of general interest to the machine learning community.

Aside from the advantage Combined EWC has over Diagonal EWC when this is indeed a useful method class, we also believe our work is of academic interest. The question around the extent to which the diagonal assumption of EWC presents a significant limitation has been outstanding in the field since its inception. Some work has made meaningful progress in answering this, as with the Kroenecker Factored FIM [4,5], however these methods do not use the full FIM and still require a fairly limiting amount of computation. Here, we found a computational tool to directly address this question. We hope that this will inspire future work, in using the SHVP and perhaps more generally in a renewed interest in EWC.  Although practitioners’ focus has shifted in recent years to competing subfields such as hierarchical RL [6] and meta-learning [7,8], we would argue that this does not preclude this work from being within the scope of ICLR or of interest to the broader ML community. In fact, one prior work which also evaluates on three atari domains aims to combine meta-learning and EWC [2]. Thus, while we do not improve upon continual learning, we improve upon the computational efficiency and general performance of EWC - answering an important question of the effect of the full FIM in the process - but also making it easier to combine EWC with more advanced techniques in the future. Hopefully, expanding the range of EWC within RL beyond Atari games.

We hope this contextualizes our work better. We are happy to discuss further on this point.

---

> ### Author Response · Authors · 2023-11-15
> **Citations for General Comment to Reviewers**
>
> [1] Kirkpatrick, James, et al. "Overcoming catastrophic forgetting in neural networks." Proceedings of the national academy of sciences 114.13 (2017): 3521-3526.\
> [2] Ribeiro, Joao, Francisco S. Melo, and Joao Dias. "Multi-task learning and catastrophic forgetting in continual reinforcement learning." arXiv preprint arXiv:1909.10008 (2019).\
> [3] Van Niekerk, Benjamin, et al. "Composing value functions in reinforcement learning." International conference on machine learning. PMLR, 2019.\
> [4] Ritter, Hippolyt, Aleksandar Botev, and David Barber. "A scalable laplace approximation for neural networks." 6th International Conference on Learning Representations, ICLR 2018-Conference Track Proceedings. Vol. 6. International Conference on Representation Learning, 2018.\
> [5] McInerney, Denis Jered, et al. "Kronecker factorization for preventing catastrophic forgetting in large-scale medical entity linking." arXiv preprint arXiv:2111.06012 (2021).\
> [6] Khetarpal, Khimya, et al. "Towards continual reinforcement learning: A review and perspectives." Journal of Artificial Intelligence Research 75 (2022): 1401-1476.\
> [7] Gupta, Gunshi, Karmesh Yadav, and Liam Paull. "Look-ahead meta learning for continual learning." Advances in Neural Information Processing Systems 33 (2020): 11588-11598.\
> [8] Javed, Khurram, and Martha White. "Meta-learning representations for continual learning." Advances in neural information processing systems 32 (2019).